# The Role of Social Media in Raising Public Health Awareness during the Pandemic COVID-19: An International Comparative Study

Mohammad Yousef Abuhashesh [1,*], Hani Al-Dmour [2], Ra'ed Masa'deh [2], Amer Salman [2], Rand Al-Dmour [2], Monika Boguszewicz-Kreft [3] and Qout Nidal AlAmaireh [2]

[1] E-Marketing and Social Media Department, Princess Sumaya University for Technology (PSUT), P.O. Box 1438, Al-Jubaiha, Amman 11941, Jordan
[2] Marketing Department, The University of Jordan, P.O. Box 1438, Al-Jubaiha, Amman 11941, Jordan; dmourh@ju.edu.jo (H.A.-D.); r.masadeh@ju.edu.jo (R.M.); asalman@ju.edu.jo (A.S.); Rand.aldmour@ju.edu.jo (R.A.-D.); Qout.nidal@gmail.com (Q.N.A.)
[3] Marketing Department, WSB University in Gdańsk, Aleja Grunwaldzka 238A, 80-266 Gdańsk, Poland; mboguszewicz@wsb.gda.pl
[*] Correspondence: m.abuhashesh@psut.edu.jo

**Abstract:** The main objective of this research is to investigate the role of social media campaigns (the type of social media platform, type of message, and message source sender) in raising public health awareness and behavioral change during (COVID-19) as a global pandemic across national selected countries (Poland and Jordan). The research utilizes a quantitative method with an exploratory and descriptive design to accumulate the initial data from a research survey given to the respondents from Jordan and Poland. A total of 1149 web questionnaires were collected from respondents in the two countries (Poland 531 and Jordan 618). In addition, multiple regression analysis was used to test the study hypotheses. The findings showed positive relationships between the components of a social media campaign, public health awareness, and behavioral change during (COVID-19) in the two countries at the same time. However, the preferred type of social media platforms, the message types and type of source sender significantly differ among the respondents due to their countries. This is the first study that examines the role of social media campaigns (the type of social media platform, type of message and message source sender) in public health awareness and behavioral change during (COVID-19) as a global pandemic in across national selected countries (Poland and Jordan).

**Keywords:** social media platform; message types; message sources; public health awareness; behavioral change

## 1. Introduction

The coronavirus 2019 pandemic (COVID-19) has become a severe public health issue worldwide, causing severe acute respiratory syndrome since it emerged from China. Therefore, it is of utmost importance to prevent any further spread of the pandemic in public and healthcare settings. Several countries have reacted according to its perception of threat, economy, healthcare policy and the healthcare system structure. They have adopted several social distancing measures, testing every suspected case, staying home, avoiding social gatherings, treating patients and contact tracing [1]. Some countries, such as Poland and Jordan have taken stringent measures to contain the pandemic outbreak, such as lockdown and mass testing. Several social media and websites have published health information about COVID-19 and have given different instructions to their users about ways to prevent the virus's spread, such as keeping a distance between themselves and others, using masks, and washing their hands. Public awareness and prevention of the coronavirus (COVID-19) infection diseases play an important role in disease control; a lack of proper knowledge of infectious diseases leads to low detection rates [2]. After COVID-19 appeared and was

transmitted to other countries outside of Mainland China, people turned to social media to know more about the virus. According to Molla [3] in just 24 h, there were 19 million mentions of COVID-19 across social media and news sites worldwide. Therefore, assessing the role of social media platforms for public awareness is critical because it helps determine the impact of prevention efforts and measures made by the government and gauges the need for intervention [4].

Merchant and Lurie [5] found that due to social media development, many methods of communicating and disseminating information and news are more available to the public. These are fast and effective and can spread the correct information as well as misinformation. According to Yigitcanlar et al. [6], health authorities can understand community perceptions and needs in a pandemic situation through social media analytics. In addition, La et al. [7] said that many countries did not circulate information about the COVID-19 outbreak or could not provide the public with the information they needed; thus, people relied on the information they could find on social media. Further, according to Gluskin et al. [8], social media platforms provide beneficial climate and socio-economic values. Furthermore, Laranjo et al. [9] indicated that social media platforms represent an essential communication source that enables creating and disseminating information to the people through the Internet. It is worth mentioning here that social media platforms allow groups and individuals to exchange information about all subjects or issues, even when they belong to minorities or are unknowns who have no chance to express their opinions using other information sources. Dubey et al. [10] argued that health information and perspectives pertinent to human health issues are revealed informally using social media platforms away from medical officials and health departments.

Despite the growing body of literature examining social media in health contexts [5,7] the authors of this study noted a lack of research on the role of social media campaigns on public health awareness and behavioral change related to COVID-19 pandemic across national countries. The critical question of this study is to explore whether the role of a social media campaign (the type of platforms, preferred message design/types, and preferred source of message) in public health awareness and behavior change during COVID-19 as a global pandemic differ across national countries. Two geographically distant and culturally different countries were selected for the analysis: Jordan and Poland. Both have a similar percentage of users concerning the entire population (2020) and a similar growth dynamic (between April 2019 and January 2020), with Jordan at 56% and 7.4%, respectively, and Poland at 50% and 7.8%, respectively [11]. Specifically, this study strives to answer the following questions:

1. Does the social media campaign (the type of platforms, preferred message design/types, and preferred source of message) have any significant relationship with public health awareness and behavioral changes during the COVID-19 pandemic across national countries (Jordan and Poland)?
2. Does the impact of a social media campaign (the type of platforms, preferred message design/types, and preferred source of message) in public health awareness and behavioral change during COVID-19 pandemic differ across national countries (Jordan vs. Poland)?

The findings of this study are expected to be helpful and essential for public health authorities and governments over the world to understand who receives the intervention (message), what impact social media platform campaigns have and what extent of changes in public health behavior and health outcomes can be attributed to the intervention, in addition to knowing how the disseminated information is perceived.

## 2. Theoretical Background

### 2.1. Social Media and Public Health

Social media is determined as a web networking that permits reach, format, and interchange of content that is universally available [12]. Social media networks can facilitate people's communication to interchange information. It has altered the method people

interact with each other, also including health conversations [13]. According to Dredze [14], people can share information and search for a topic on Twitter, so people can tweet health topics over the public health community to get specific advice or solutions. The healthcare sector, covering medical centers, health regulations, hospitals, professional associations, pharmaceutical corporations, and patient support groups, employs social media for numerous objectives. It can assist healthcare experts and patients' communications regarding health cases that can bring new changes and challenges to the health care sector [15].

Health content on social media platforms lately has been progressively sent from devices and mobile phones, which supports the opportunity to carry geographic data [16]. Medical descriptions are more beneficial, and social media tools permit data to be presented in various forms. For example, videos on YouTube can be adopted as a substitute for written messages in the fields with low knowledge rates, where individuals cannot read medical records [17]. Likewise, a variation of other forms of social media is applicable to motive debating between patients and comfort specialists [18,19]. These channels consist of Twitter, Myspace, Blogs, Wikipedia, and Email, besides Facebook that is managed on an extensive measure to spread medical materials referred to epidemics, clinics, drugstores, and examination by the diseased person and health specialists [20]. In addition to companies in the private sector, health concern agencies and public health institutions locally and nationally and are using Twitter as a prime electronic platform for the education and enhancement of health, as the more significant part of the content on Twitter is overtly ready and may provide an origin for health-concerning information [21]. Twitter is a free network on social media with over 100 million energetic users throughout the day. Despite its extensive adoption by healthcare experts and professionals, there are few studies on the shapes of hashtag use on Twitter and a lack of manifest research concerning its influence on public health. In [22], Seymour stated that planned social media use, such as hashtags, may be an efficient method for organizations to transmit precise information to the audience through a period of crisis [23]. Users may be guided to links and pursue private health foundations and service providers for accurate geographic-related information. Examiners may utilize general available "big data" such as local tweets to measure the hazard interaction efforts of native agencies [23]. For example, public authorities were able to track people's sentiment toward the Zika virus's breakout through the analysis of Yahoo! Answers. Public reactions became less aggressive as the government published more answers about virus prevention and treatment [24].

Public health is "the discipline and profession of avoiding infection and diseases, continuing existence and improving individual health via coordinated intentions and educated preferences of communities, institutions, general and private societies, as well as humans" [25]. Previous researchers Korda and Itani [26] have investigated the implications and effectiveness of utilizing social and digital media to promote public health and disease prohibition efforts. So, the platforms of social media assist people in connecting to health-encouraging messages, and the content of the message itself can create a public that is mindful of their health conduct. If the public is motivated and start spreading these messages through social media channels, then this behavior shape general ratifies the health attitude. Consequently, sharing health messages through social networking channels matches two primary standards: alteration of conduct and general obligation [27]. Over the engagement of social media platforms, public health foundations can share related content anywhere users spend their time [28].

Additionally, studies have shown that digital advertisements can be a useful tool for campaigns to extend to niche viewers, who are the receivers of the content. Digital and social media ads can strictly objective inhabitances according to their features such as geographical areas, gender, age, and interests. They can increment campaign range and are the most excellent strategy to transfer communications to the most significant amount of the concerned audience. Because ads can be purposely placed, and views can be ensured, the prime measurement of the success in ads is the reached public [29]. According to the researchers, George et al. [30] assume that social media has an immediate

public health relation because media networks could substantially affect health conduct and outputs. However, organizations related to public health have not yet utilized social media's real chances [31]. Chou et al. [32] mainly investigated the necessity for public health's interference to use social media's interaction environment. Heldman et al. [33] proposed that foundations of public health and practitioners frequently utilized social media for the conventional one-way transmission of information; instead of engaging audiences in modern ways for interaction such as two-way communication, or what is called "indeed social".

Social media could encourage healthy lifestyles to increment public health awareness via exercises and healthy diets [34]. For this reason, public health awareness activities are prepared every year to update the information of health hazards and interactions during a particular period, which could be for days, weeks or months. The amount of public awareness campaigns has significantly increased [35]. Social media can collect information concerning hospital and doctor's performance based on their patients' experiences [36]. It is also applied for health progress and health studies. Furthermore, it can be noticed that sharing healthy materials and information such as learning from other people and increasing health education and knowledge are primary reasons for which patients adopt social media platforms and the web referred to healthcare [37]. Thus, a significant investigation is that electronic campaigns can attain great reach at a relatively minimum cost [38].

Healthcare can be perceived and better comprehended by patients if they have health-generated content on social media [39]. Individuals would rather have electronic information about health over the device they are most usual with, according to what was investigated in an Australian survey [40]. Notably, youthful people favor obtaining health information via the Internet or by social networking means; however, older adults favor newspapers. Hereafter, these preferences would be directed through online choices towards a scope of age categories. Nowadays, a different method is suggested according to previous evaluation of social preferences. Most studies on preliminary care preparation include people above 65 years; a new direction is now implemented toward including and instructing many younger people; therefore, they are more ready to interact with these issues in their societies and families. One study looks to students at US universities and suggests that an essential part of public health is issuing credible information related to preliminary care preparation to youthful people [41]. Additionally, over half of US grownups in all ethnic and age categories use the web to look for health materials and information [42].

*2.2. Social Media and Public Health Awareness of Infectious Diseases*

Infectious diseases are the contagious diseases generated by various kinds of infective micro-organisms, such as microbes, germs, viruses, and parasites, and these forms of infections can be transmitted instantly or indirectly from one human to another [43]. The World Health Organization (2020) determined the outbreak of COVID-19 on the 30th of January in 2020, stating that is was a "public health emergency of international concern". Scholars quickly began working to explain the pandemic characteristics, covering its ability to transmit, death rate, and emergence [44]. Both the World Health Organization (WHO) and offices for disease surveillance and prohibition provide regular communications through social media and website upgrade networks. At the time of the crisis of public health, it is crucial to communicate instant information to the mass population in the actual period while tempering media subjection that can drive to traumatic stress reactions and related diseases. As authenticated society agents, health care experts also play an essential part in contacting necessary information to ill people and other society members. Workable guidance that persons can apply to be safe from infectious microbes and viruses include the cleaning of hands, the use and instant disposal of hankies for sneezing and coughs, the sterilization of surfaces, and social distancing are all beneficial, while at the same moment working to block other mutual infections such as influenza [45].

Therefore, publicists and public health administrators worked to transfer crucial information worldwide related to risk valuation and suggestions, and concerned warnings appeared regarding the psychosomatic suffering generated from repeated media subjection to the outbreak. This has included the instant suffering in the inhabitants previously struggling with unmatched economic and social collapse and the downstream impacts on mental and physical health [45]. Furthermore, these tension responses can increment help-searching conduct that may be odd or not endorsed in answer to the current risk, overloading health concerns facilities and redirecting crucial resources. For example, a phobia of purchasing primary consumers materials such as first-aid packages, toilet paper, bottled water, alcohol and chlorine sprays, and hand sanitizer in reaction to COVID-19 has driven to worldwide shortages price leveraging of substantial requirements [45]. Throughout the health emergency, the audience relies on media channels to transfer precise and modern information to make knowledgeable decisions about health defensing behaviors. Through crises and risks, the dependence on media will be increased and the sources must be credible and accessible to provide hazards appraisals and recommendations [23].

Social media are playing an active role in transmitting prompt caution signs to public health officials to make an update path of work to prohibit and monitor the spreading of those diseases [46]. It is also a helpful tool for the active interaction for the new updates regarding infectious disease's news and primary medical details to the audience. These kinds of diseases appear to have hazards to the public, who frequently switch to conventional media and social media to obtain information [47]. So that the audience understands the risks that can be shaped and constituted by describing and communicating about infectious diseases via social network channels, which can have a significant influence on their behaviors and decision making, Patel et al. [40] advise utilizing social media to support different areas: emotional, empirical, and social.

Several investigators have discovered increasing the conventional alerting networks regarding the disease occurrence with statistics exported from Twitter. Through manually investigating a massive number of tweets, which presented that reports coming from individuals themselves about their symptoms are the utmost credible sign in examining if a tweet is related to an outbreak or not, new trends of diseases and infection's outbreaks are studied and captured by scholars, often influenza, by observing social media channels regarding infectious diseases [48]. Other investigators concentrate on new elaborate designing of the tweet's style and language and their relation to public health in common and to forecast influenza specifically [49]. Furthermore, wealthy contents can be utilized to find out local health update movements, making digital monitoring of contagious diseases reasonable [50].

Suitable time to identify the flare-up of the infectious disease is a sensitive issue, for both active starting of public health interference and surveillance actions, in addition to the appropriate time to notifying government organizations and audiences at massive. The observation ability for such revelation can be costly; public health infrastructures in many countries have a shortage to determine an outbreak at its advanced stages. Web networking is changing how pests and epidemic intelligence work, and it suggests a resolution to these limitations [46]. Governments universally are starting to utilize the web and information technology in an effort to direct citizens' needs for more important information available, transparency, and gain entry to general services. One network during which the previous goals are being followed is social media, which cover connecting sites, such as Twitter, YouTube, and Facebook [51].

### 2.3. Social Media and Public Health Behavioral Changes

The diffusion of usage of social media channels can as well impact the behaviors related to public health and objectives through the social build-up since human beings are very social, they are frequently affected by their families, as well as by friends, and by mutual friends [47]. The campaigns generated by social media can immediately or indirectly impact behaviors related to health towards considerable inhabitants. Thus,

conscious planning of the subjects and the materials of campaigns, and their match with the target audiences are essential. They are perceived as a reliable way of preventing diseases and the efficiency of its costs [52]. Research on social marketing campaigns, involving Internet campaigns, proposes that they can impact people to alter their conduct and attitudes [53].

A substantial constraint for social media campaigns' success is that humans should sense trust in the materials and information given. An example shows that an appraisal for TV ads in Canada regarding health improvement pointed at older adults showed that people who receive were mostly doubting the data if they recognized that it was sent by the government. Experts such as doctors or famous persons were viewed as more reliable. The outstanding benefit of applying social media in healthcare is the understandable accessibility of material linked to infections, diseases, treatment strategies, and remedies [54]. A powerful incentive is emotional support provided by connecting with humans influenced by the same illness and simplifying the diffusing of updates regarding the newest medications. This connection is not restricted between patients; it enhances the relationship between patients and healthcare experts. In an expression of price, the area of a specific facility, and the remediation excellence, it will assist in contrasting diverse services suppliers in the attainable zone [54].

Social scholars have examined that media channels and technology have a further immediate impact as being an orderly part of youth individual's social and behavioral experience. It has been suggested that technology and media are now a part of the environmental framework of human beings. Keeping in mind the relevance of environmental elements on health behavior, it would be rational to check the relation between youth mature social media usage and health conduct [55]. New technologies and media such as movies, Internet and video games are conceived as stationary activities that replace exercise, which over time lead to increases in the body mass index (BMI) of children and youth [56]. With technological interchange and the capability to reach diverse technologies from a single device, a considerable number of youth have gained access to social media channels on their mobile phone and can choose to use them at any time, which could be a signal for an increment in passive behavior. The interactive pattern of social networking sites may have various effects on youth and their health behavior, more so than unresponsive technologies such as movies, TV, and music. Investigators have determined that social participation can influence food preferences and selections amongst young people [57].

Therefore, examining messages and conduct on social media might give a special vision for the potential impacts of information concerning health connecting vast audience segments [31]. Data circulation on social media can profoundly impact people conduct and attitudes and change the efficiency of the countermeasures installed by governments. For this reason, patterns that predict virus diffusing begin to consider behavioral reactions of the public about general health involvement and the interaction drivers behind consuming the content [58]. The short interference by campaigns on social media regarding public health has been seen to have a significant influence on optional conduct change [38]. As solid proof indicates that public health promotion campaigns managed via social media can have an immediate and positive impact on behavior [52].

Furthermore, differences within generations in social media use are also an important factor in changing behavior toward health. Web users' choices and the usage of electronic channels also vary by ethnicity, gender, income, and education features. Women are more probable than men to look for information regarding symptoms, remediation, infections, diseases, and drugs. Men to a greater extent are more likely than women to search objectively for supplements and vitamins, agencies for health insurance, and medical practitioners; both women and men were equally likely to search for wellness and healthy subjects; and men are more likely to utilize social media than women. Therefore, developmental studies concentrated on comprehending the interests, essential values, and desires of the public that assist the content to be culturally applicable and sense creditable. The results assessment will assess the differences in attitudes, conduct, knowledge, and

beliefs to define behavioral actions variations. The feeling of empowerment in deciding somebody's health can be an essential part of assisting individuals in looking at lifestyles and good health behavior. Various studies have noted that internet interference had a considerable positive impact on empowerment. A new examination to measure the efficiency of web interference in raising patients' empowerment contrasted with regular care by face-to-face interventions found a significant impact on empowerment as valued on self-efficiency extent and scales of proficiency [59]. Other studies appear that patients would experience their health decisions through online learning, discovering they have increment confidence in asking health providers questions and information to assist them to run their situations [60].

Global mandates to promote public health, depending on modifications in an extensive collection of behaviors, such as bodily activities and nutrition programs, to smoking and substance use to the commitment to therapy and examination rules. Consequently, attempts to match these mandates rely on interference procedures efficiently amending the audience's behavior to public health [61]. An example of health awareness campaigns are that an entire campaign was formed that contains the Fresh Empire website's foundation as a health brand. Health awareness is a plan that utilizes marketing rules to promote substantial behavior change [62]. To create the brand and transfer the campaign content, the master plan incorporated various messaging platforms to smooth the range and interaction. The content channel for Fresh Empire contains paid and organic social media, digital ads, broadcast and a website, plus events. To engage the teenagers on the networks they use generally, there is an intensive digital concentration.

## 3. Research Model and Hypotheses

According to the health belief model, people tend to take preventive actions if they feel seriously threatened. These health interventions should address the specific perceptions of individuals about susceptibility and benefits [9]. The behavioral change approach has improved health by changing people's lifestyles [63]. It is assumed that individuals must understand basic facts about a particular issue related to their health to change their lifestyles as a result of feeling threatened, especially by an infectious disease. In this context, individuals should learn a group of skills and have access to timely services. Behavior changes include, but are not limited to hand washing, wearing masks, social distancing, avoiding public gatherings, sanitation, and isolation. Interventions aiming at promoting public health can improve health quality in the society and support the policies and programs run by official health authorities in fighting the outbreak and spread of infectious diseases. If people trust these policies and programs, they are likely to respond positively to public health interventions and participate in the launched health promotion programs in large numbers. Social media health campaigns can induce positive behavioral changes and even eliminate negative ones in individuals. According to Laranjo et al. [9] among the advantages of social media health interventions are cost-efficiency, ubiquity and passing geographical barriers. The tremendous growth in social media networking sites opens the door wide to more opportunities to disseminate health interventions to the general public in real-time and irrespective of geographical location, thus leading to public health promotion and positive behavioral changes.

Therefore, an integrated conceptual model was developed to guide the current study objectives. It is assumed that the use of social media platforms, the preferred type of message and source of message would increase public protection and prevention against coronavirus (COVID-19) disease across national countries, and they are varied among the respondents' nationality, i.e., Jordan vs. Poland. In this study, the variables of primary interest (the independent variables) are the social media platforms, the type of preferred message and type of source of the message. The dependent variable (public awareness, behavioral change, and public protection against the coronavirus (COVID-19) disease) and the moderating variables are the respondents' nationality. The expected relationships among these variables are illustrated in Figure 1.

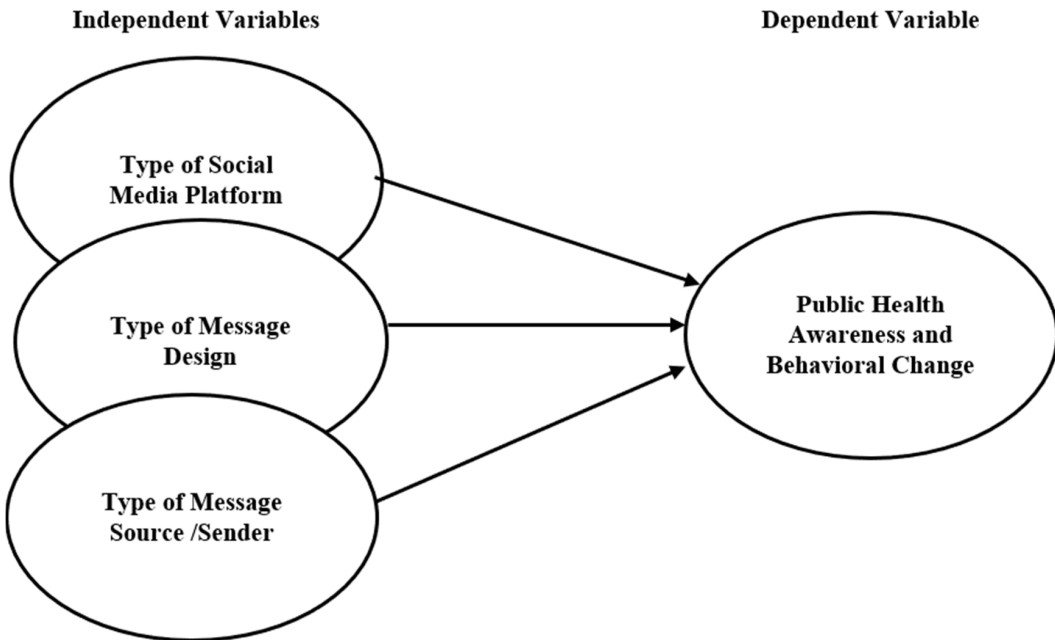

**Figure 1.** The study model.

Based upon the above arguments, the following major two hypotheses were formulated concerning the role of mass media campaigns in raising public awareness of the coronavirus (COVID-19) as pandemic disease.

**Hypothesis 1.** *The social media campaign (the type of platforms, preferred message design/types, and preferred source of message) do not have any significant relationship with public health awareness and behavior change during COVID-19 pandemic across national selected countries (Poland and Jordan).*

This first hypothesis is divided into three sub hypotheses:

**Hypothesis 1a.** *The type of media social platforms will not significantly impact public health awareness and behavioral change during the COVID-19 pandemic across national countries.*

**Hypothesis 1b.** *The type of social media preferred message will not significantly impact public health awareness and behavioral change during the COVID-19 pandemic across national countries.*

**Hypothesis 1c.** *The type of social media preferred messaging source will not significantly impact public health awareness and behavioral change during the COVID-19 pandemic across national countries.*

**Hypothesis 2.** *There is a significant difference on the impact of the social media campaign (the type of platforms, preferred message design/types, and preferred source of message) in public health awareness and behavior change during the COVID-19 pandemic due to the respondents' country (Poland vs. Jordan).*

This second hypothesis is divided into three sub hypotheses:

**Hypothesis 2a.** *There is a significant difference in the impact of social media platforms in public health awareness and behavioral change during the COVID-19 pandemic due to the respondents' country (Poland vs. Jordan).*

**Hypothesis 2b.** *There is a significant difference in the impact of social media's preferred message in public health awareness and behavioral change during the COVID-19 pandemic due to the respondents' country (Poland vs. Jordan).*

**Hypothesis 2c.** *There is a significant difference in the impact of the type of social media preferred source in public health awareness and behavioral changes during COVID-19 pandemic due to the respondents' country (Poland vs. Jordan).*

## 4. Methodology

The research utilizes a quantitative method with an exploratory and descriptive design. To confirm the research's conceptual model and to examine the research hypotheses, a survey questionnaire was executed to collect it. This research's target population consisted of all users of mass media channels in Jordan and Poland. A convenience sampling technique was employed. Consequently, a sample size of 1149 respondents was reached. The questionnaire consisted of two sections; the first section in the questionnaire presents general personal information about a respondent, gender, age, educational level, and country. The second section includes questions to measure the independent and dependent variables. The questionnaire's content was mainly selected and adopted from relevant previous studies [64] using 5-points Likert scale ranging from strongly disagree = 1 to strongly agree = 5 to measure four independent variables: social media network, public awareness, behavioral change, and public protection. In contrast, the dependent variables, namely type of message appeal, and source of the message, were measured using-points on a Likert scale ranging from not necessary at all = 1 to very important = 5. A web link for the online questionnaire was sent out to the potential respondents between 15 May to 30 November in 2020.

### 4.1. Respondents Demographic Profile

As indicated in Table 1, 52.2% of the respondents were females, and 52.1% were in the age group between 18 and 34. Additionally, 44.9% of the respondents have a bachelor's degree (undergraduate) in both countries.

**Table 1.** Sample profile (*N* = 1149).

| Category | Category | Jordan | % | Poland | % | All | % |
|---|---|---|---|---|---|---|---|
| Gender | Male | 322 | 52.1 | 216 | 40.6 | 538 | 46.8 |
| | Female | 296 | 47.9 | 315 | 59.1 | 611 | 52.2 |
| | Total | 618 | | 531 | | 1149 | |
| Age | 18 to ≤34 years | 275 | 44.4 | 324 | 60.1 | 599 | 52.1 |
| | 34 to ≤54 years | 215 | 34.7 | 131 | 24.6 | 346 | 30.1 |
| | 54 and over | 128 | 20.7 | 76 | 14.3 | 204 | 17.7 |
| | Total | 618 | | 531 | | 1149 | |
| Educational level | High school and Diploma | 167 | 27.0 | 133 | 25.0 | 300 | 26.1 |
| | Undergraduate | 297 | 48.0 | 256 | 48.2 | 517 | 44.9 |
| | Graduate | 154 | 24.9 | 142 | 26.7 | 332 | 28.9 |
| | Total | 618 | | 531 | | 1149 | |

### 4.2. Data Analysis

Reliability and validity analyses were conducted; descriptive analysis was used to describe the sample's characteristics and the respondent to the questionnaires besides the independent and dependent variables. Multiple regression analysis and ANOVA tests were employed to test the research hypotheses. Cronbach's alpha coefficient measured the reliability of the instrument. Further, some scholars (e.g., Bagozzi and Yi, [65]) suggested that all indicators or dimensional scales should be above the recommended value of 0.60. Table 2 represents the results of Cronbach's alpha for the independent and dependent variables. Cronbach's alpha coefficients of all the tested variables are above 0.60, suggesting the composite measure is reliable.

**Table 2.** The Cronbach's alpha coefficients of study variables.

| Variables | Number of Items | Cronbach Alpha |
|---|---|---|
| Social Media Platforms | 5 | 0.820 |
| Type of Message design | 6 | 0.781 |
| Source of the Message/sender | 5 | 0.848 |
| Public Awareness and Behavioral Change | 4 | 0.896 |

*4.3. Descriptive Analysis*

To illustrate the respondents' attitudes toward each question, they were asked in the assessment, and accordingly, the mean was calculated for all the measurements. The descriptive statistics offered in Table 3 pointed to a positive disposition towards the items measured in both countries. Additionally, the items were being ordered by their means [66]. Table 2 answers the research questions and shows that social media campaigns (the type of platforms, preferred message design/types, and preferred message source/sender) effectively employed public health awareness during COVID-19 in both countries (Jordan and Poland). It is worthy of indicating that Facebook is the most popular channel in both countries. It also shows that TAM1 (conversational appeal) is the most crucial message type that is positively associated with the public health awareness in both countries followed by TAM2, TAM4, TAM5 and TAM6 found to be applied to a moderate level of importance. Additionally, Table 3 shows that SMG1 (physicians/sender) is the most crucial source of the message that is positively associated with the public health awareness in both countries, followed by SMG3, SMG4, while SMG2 and SMG5 were applied to high levels of importance.

**Table 3.** The mean and standard deviation of the study variables.

| Code | Constructs | Jordan | | Poland | | Both | |
|---|---|---|---|---|---|---|---|
| | **Mass Media Channels** | **Mean** | **Rank** | **Mean** | **Rank** | **Mean** | **Rank** |
| SOM1 | Facebook | 3.71 | 1 | 3.56 | 1 | 3.64 | 1 |
| SOM2 | Twitter | 3.69 | 2 | 3.13 | 3 | 3.41 | 2 |
| SOM3 | YouTube | 3.29 | 3 | 2.81 | 4 | 3.05 | 4 |
| SOM4 | Instagram | 3.11 | 4 | 3.31 | 2 | 3.21 | 3 |
| SOM5 | Other | 2.98 | 5 | 2.41 | 5 | 2.695 | 5 |
| | Type of Message design/Appeal | | | | | | |
| TAM1 | Conversational appeal | 4.16 | 1 | 3.69 | 2 | 3.94 | 1 |
| TAM2 | Scientific descriptive | 3.99 | 3 | 3.83 | 1 | 3.92 | 2 |
| TAM3 | Anecdotal appeal. | 3.66 | 4 | 3.23 | 4 | 3.46 | 4 |
| TAM4 | Martyrdom appeal | 4.09 | 2 | 3.32 | 3 | 3.74 | 3 |
| TAM5 | Cartoons panic appeal. | 3.57 | 5 | 2.64 | 6 | 3.14 | 5 |
| TAM6 | Humorous appeal | 3.02 | 6 | 2.92 | 5 | 2.98 | 6 |
| | Source of the Message | | | | | | |
| SMG1 | Physicians. | 4.31 | 2 | 3.76 | 2 | 4.03 | 2 |
| SMG2 | Academics and experts. | 3.69 | 4 | 3.44 | 3 | 3.57 | 5 |
| SMG3 | Media health professionals/reporters | 3.88 | 5 | 4.11 | 1 | 3.98 | 2 |
| SMG4 | Public Security agencies. | 4.21 | 3 | 3.42 | 4 | 3.85 | 3 |
| SMG5 | Senior State health officials | 4.48 | 1 | 3.16 | 5 | 3.82 | 4 |

Multiple regression analysis was used to test the study hypotheses. Furthermore, normality of the independent variables and the absence of multi co-linearity problem (a case of multiple regression in which the independent variables are themselves highly correlated) were checked. According to Hair et al. [67] most of the variance inflation factor (VIF) values should be less than 10. VIF is the reciprocal of the tolerance value; small VIF values show a low correlation among the variables. For this purpose, tolerance, and variance inflation (VIF) were investigated; Table 4 includes the results.

**Table 4.** Tolerance and VIF for the independent variables.

| Variables | Tolerance | VIF |
|---|---|---|
| Social media platforms | 0.224 | 4.465 |
| Massage type | 0.156 | 6.415 |
| Message sources | 0.182 | 5.506 |
| Public awareness and behavioral change | 0.345 | 2.899 |

As shown in Table 4, the VIF values were less than the critical value (10), which is most common among the most studies, suggesting no multi-linearity problem among the independent variables.

Table 5 shows that the multiple correlation coefficient, R = 0.784, indicates a positive correlation between the independent variables (the type of social media platform) and public awareness and behavioral change in both countries. The $R^2$ indicated the generalizability of the model. It allows us to generalize the results taken from the respondents to the whole population. In this case, it equals 0.614. The results showed that F-ratio for these data is equal to 159.017, which is statistically significant at $p \leq 0.05$. Therefore, we conclude that there is a statistically significant impact of the type of social media platforms as independent variables on the public awareness in the two countries. The β indicates each predictor's contribution (independent variable) to the model if other predictors are held constant. Table 5 shows the standardized coefficients for each independent variable. The values of β for SOM1, SOM2, SOM3, MMC4 and SOM5 are 0.254, 0.239, 0.179, 0.160 and 0.192 respectively, which are positive. We can infer from the values of beta that the variables with the highest contribution in the model are SOM1, SOM2, SOM3, MMC4 and SOM5. Accordingly, the hypothesis (H1a) was rejected.

**Table 5.** Result for hypothesis (H1a).

| Variables (Type of Social Media Platforms) | | R | $R^2$ | F | Sig (f) | B | T | Sig (t) |
|---|---|---|---|---|---|---|---|---|
| Constant | | | | | | | 11.885 | 0.000 |
| SOM1 | Facebook | 0.784 | 0.614 | 0.612 | 0.000 | 0.254 | 10.962 | 0.000 |
| SOM2 | Twitter | | | | | 0.239 | 10.314 | 0.000 |
| SOM3 | YouTube | | | | | 0.179 | 7.487 | 0.000 |
| SOM4 | Instagram | | | | | 0.160 | 6.338 | 0.000 |
| SOM5 | Other | | | | | 0.192 | 7.852 | 0.000 |

Predictors: (constant), social media platforms: SOM1. SOM2, SOM3, SOM4 AND SOM5 Dependent variable: public awareness and behavioral change.

Table 6 shows that the multiple correlation coefficient R = 0.505 indicates a positive correlation between the preferred message type and public health awareness and behavioral change in the WO countries. $R^2$ equals to 0.255. The results showed that F-ratio for these data is equal to 194.421, which is statistically significant at $p \leq 0.05$. Consequently, we conclude a statistically significant impact of the independent variables (preferred message type) in the public health awareness and behavioral change in both countries. We can infer

from the values of beta that the variables with the highest contribution in the model are only TAM1, TAM2 TAM3, and TAM4, accordingly, hypothesis (H1b) was rejected.

**Table 6.** Result for the (H1b) hypothesis.

| Variables (the Type of Preferred Message Design) | | R | $R^2$ | F | Sig (f) | B | T | Sig (t) |
|---|---|---|---|---|---|---|---|---|
| Constant | | | | | | | 9.020 | 0.000 |
| TAM1 | Conversational appeal | 0.505 | 0.255 | 0.251 | 0.000 | 0.120 | 3.595 | 0.000 |
| TAM2 | Scientific descriptive appeal | | | | | 0.175 | 5.157 | 0.000 |
| TAM3 | Anecdotal appeal. | | | | | 0.262 | 7.917 | 0.000 |
| TAM4 | Martyrdom appeal | | | | | 0.098 | 3.078 | 0.002 |
| TAM5 | Cartoons panic appeal. | | | | | 0.023 | 0.811 | 0.418 |
| TAM6 | Humorous appeal | | | | | 0.035 | 1.148 | 0.251 |

Predictors: (constant), preferred message: TAM1, TAM2, TAM3, TAM4, TAM5, TAM6. Dependent variable: public awareness and behavioral change.

Table 7 shows that the multiple correlation coefficient R = 0.463 indicates a positive correlation between the independent variables (the type of message source) and public health awareness and behavioral change in both countries. $R^2$ equals to 0.214. The results showed that F-ratio for these data is equal to 251.241, which is statistically significant at $p \leq 0.05$. Consequently, we conclude a statistically significant impact of the type of message source in the public health awareness and behavioral change in both countries. We can infer from the values of beta that the variables with the highest contribution in the model are SMG1, SMG3, SMG4 and SMG5. Accordingly, the hypothesis (H1c) was rejected.

**Table 7.** Result for (H1c) hypothesis.

| Variables (Message Source) | | R | $R^2$ | F | Sig (f) | B | T | Sig (t) |
|---|---|---|---|---|---|---|---|---|
| Constant | | | | | | | 9.355 | 0.000 |
| SMG1 | Physicians. | 0.463 | 0.214 | 0.211 | 0.000 | 0.134 | 3.847 | 0.000 |
| SMG2 | Academics, professors, and experts. | | | | | 0.005 | 0.172 | 0.864 |
| SMG3 | Media health professionals/reporters | | | | | 0.157 | 4.844 | 0.000 |
| SMG4 | Public Security agencies. | | | | | 0.126 | 3.153 | 0.002 |
| SMG5 | Senior State health officials | | | | | 0.154 | 3.916 | 0.000 |

Predictors: (constant), message source sender: SMG1, SMG2, SMG3, SMG4 and SMG5. Dependent variable: public awareness and behavioral change.

ANOVA tests were also employed to test the sub hypotheses derived from the 2nd primary hypothesis (H2a, H2b, and H2c). It was used to assess whether the impact of social media campaigns (the type of social media platform, message types, and message sources) on public health awareness and behavioral change differ among the study's respondents across national countries during COVID-19 pandemic (Poland vs. Jordan). Results of ANOVA tests for theses hypothesis are summarized in Tables 8–10. The results indicated significant differences among the respondents' country (Poland vs. Jordan) in terms of their perceptions of the role of social media campaigns in public health awareness and behavioral change during COVID-19 as a global pandemic.

**Table 8.** ANOVA analysis result for hypothesis (H2a).

| Type of Social Media Platform | | Sum of Squares | Df | Mean Square | F | Sig. |
|---|---|---|---|---|---|---|
| SOM1 | Between Groups | 211.183 | 1 | 211.183 | 153.443 | 0.000 |
| | Within Groups | 1578.615 | 1147 | 1.376 | | |
| | Total | 1789.798 | 1148 | | | |
| SOM2 | Between Groups | 92.804 | 1 | 92.804 | 70.018 | 0.000 |
| | Within Groups | 1520.268 | 1147 | 1.325 | | |
| | Total | 1613.072 | 1148 | | | |
| SOM3 | Between Groups | 122.651 | 1 | 122.651 | 89.470 | 0.000 |
| | Within Groups | 1572.379 | 1147 | 1.371 | | |
| | Total | 1695.030 | 1148 | | | |
| SOM4 | Between Groups | 446.363 | 1 | 446.363 | 287.680 | 0.000 |
| | Within Groups | 1779.676 | 1147 | 1.552 | | |
| | Total | 2226.038 | 1148 | | | |
| SOM5 | Between Groups | 119.503 | 1 | 119.503 | 77.263 | 0.000 |
| | Within Groups | 1774.060 | 1147 | 1.547 | | |
| | Total | 1893.563 | 1148 | | | |
| All | Between Groups | 181.169 | 1 | 181.169 | 231.245 | 0.000 |
| | Within Groups | 898.617 | 1147 | 0.783 | | |
| | Total | 1079.786 | 1148 | | | |

**Table 9.** ANOVA analysis result for hypothesis (H2b).

| Type of Message | | Sum of Squares | Df | Mean Square | F | Sig. |
|---|---|---|---|---|---|---|
| TAM1 | Between Groups | 63.847 | 1 | 63.847 | 60.361 | 0.000 |
| | Within Groups | 1213.246 | 1147 | 1.058 | | |
| | Total | 1277.093 | 1148 | | | |
| TAM2 | Between Groups | 6.666 | 1 | 6.666 | 6.440 | 0.011 |
| | Within Groups | 1187.313 | 1147 | 1.035 | | |
| | Total | 1193.979 | 1148 | | | |
| TAM3 | Between Groups | 52.454 | 1 | 52.454 | 41.517 | 0.000 |
| | Within Groups | 1449.149 | 1147 | 1.263 | | |
| | Total | 1501.603 | 1148 | | | |
| TAM4 | Between Groups | 170.134 | 1 | 170.134 | 151.217 | 0.000 |
| | Within Groups | 1290.489 | 1147 | 1.125 | | |
| | Total | 1460.623 | 1148 | | | |
| TAM5 | Between Groups | 244.780 | 1 | 244.780 | 183.067 | 0.000 |
| | Within Groups | 1533.661 | 1147 | 1.337 | | |
| | Total | 1778.440 | 1148 | | | |

**Table 9.** *Cont.*

| | Type of Message | Sum of Squares | Df | Mean Square | F | Sig. |
|---|---|---|---|---|---|---|
| | Between Groups | 2.742 | 1 | 2.742 | 1.703 | 0.192 |
| TAM6 | Within Groups | 1846.670 | 1147 | 1.610 | | |
| | Total | 1849.412 | 1148 | | | |
| | Between Groups | 64.427 | 1 | 64.427 | 116.097 | 0.000 |
| All | Within Groups | 636.513 | 1147 | 0.555 | | |
| | Total | 700.940 | 1148 | | | |

**Table 10.** ANOVA analysis result for hypothesis (H2c).

| | Type of Source | Sum of Squares | Df | Mean Square | F | Sig. |
|---|---|---|---|---|---|---|
| | Between Groups | 149.508 | 1 | 149.508 | 142.913 | 0.000 |
| SMG1 | Within Groups | 1199.928 | 1147 | 1.046 | | |
| | Total | 1349.436 | 1148 | | | |
| | Between Groups | 18.704 | 1 | 18.704 | 16.076 | 0.000 |
| SMG2 | Within Groups | 1334.478 | 1147 | 1.163 | | |
| | Total | 1353.182 | 1148 | | | |
| | Between Groups | 14.485 | 1 | 14.485 | 13.694 | 0.000 |
| SMG3 | Within Groups | 1213.233 | 1147 | 1.058 | | |
| | Total | 1227.718 | 1148 | | | |
| | Between Groups | 177.573 | 1 | 177.573 | 161.116 | 0.000 |
| SMG4 | Within Groups | 1264.160 | 1147 | 1.102 | | |
| | Total | 1441.734 | 1148 | | | |
| | Between Groups | 373.692 | 1 | 373.692 | 305.883 | 0.000 |
| SMG5 | Within Groups | 1401.271 | 1147 | 1.222 | | |
| | Total | 1774.963 | 1148 | | | |
| | Between Groups | 82.458 | 1 | 82.458 | 124.753 | 0.000 |
| All | Within Groups | 758.127 | 1147 | 0.661 | | |
| | Total | 840.584 | 1148 | | | |

## 5. Discussion

This study investigated social media campaigns in public health awareness and behavioural change during COVID-19 pandemic across national selected countries, Poland and Jordan. To achieve the study objectives and conduct the research using a systematic approach, a conceptual framework was developed based on a communication literature review and belief health change theory. The potential benefits of using social media platforms applications in public health awareness and behavioural change include disseminating public health interventions, enhancing public awareness, promoting healthy behaviour, improving health outcomes, and providing health information to the community [68]. The analysis provides empirical evidence regarding the role of social media campaigns (the type of social media platform, message types, and message sources) in public health awareness and behaviour change during COVID-19 as a global pandemic in the two countries at the same time (hypotheses H1a, H1b and H1c). These three sub-hypotheses are significantly and positively supported the linkage between social media campaigns and public health awareness and behavioural change across national countries (Poland and Jordan). These results are supported by previous studies (e.g., Al-Dmour et al. [64]).

However, the results of the second significant hypotheses and its sub-hypotheses (H2a, H2b and H2c) showed that the perception of the role of social media campaigns (the type of social media platform, message types, and message sources sender) in public health awareness and behaviour change during COVID-19 differ among respondents in terms of their country (Poland vs. Jordan). These results indicated that people in different countries respond differently to social media health campaigns toward social media platforms, message types and message sources sender, even when the pandemic is a gable disease. This might be attributed to the national culture and the media habits for people in the two countries. Even though Facebook is the most popular social media in the two countries, other social media platforms' ranking found differences between the two groups (see Tables 11 and 12).

**Table 11.** Social media platforms' ranking mean.

| Code | Constructs | Jordan | | Poland | | Both | |
| --- | --- | --- | --- | --- | --- | --- | --- |
| **Media Channels** | | **Mean** | **Rank** | **Mean** | **Rank** | **Mean** | **Rank** |
| SOM1 | Facebook | 3.71 | 1 | 3.56 | 1 | 3.64 | 1 |
| SOM2 | Twitter | 3.69 | 2 | 3.13 | 3 | 3.41 | 2 |
| SOM3 | YouTube | 3.29 | 3 | 2.81 | 4 | 3.05 | 4 |
| SOM4 | Instagram | 3.11 | 4 | 3.31 | 2 | 3.21 | 3 |
| SOM5 | Other | 2.98 | 5 | 2.41 | 5 | 2.695 | 5 |

**Table 12.** Social media platforms' ranking percentages.

| Code | Media Channels | Jordan | | Poland | | World * |
| --- | --- | --- | --- | --- | --- | --- |
| | | **%** | **Rank** | **%** | **Rank** | |
| SOM1 | Facebook | 69% | 1 | 49% | 2 | 35% |
| SOM2 | Twitter | 10% | 3 | 3.8% | 4 | 3.6% |
| SOM3 | YouTube | No data | - | No data | - | - |
| SOM4 | Instagram | 29% | 2 | 22% | 3 | 15% |
| SOM5 | | | | | | |

* https://datareportal.com/reports/digital-2019-global-digital-overview, (accessed on 18 July 2021).

The potential number of people that marketers can reach using adverts on each of platforms compared to total population aged 13+.

Comparison of the ranking of platforms indicated by respondents that are important in shaping their health awareness and behavior change during COVID-19 pandemic does not coincide (with the exception of Facebook in Jordan) with the platforms most used in their countries (assessment based on the potential number of people that marketers can reach using adverts on each of them). The data obtained in this study do not justify the conclusion that in crisis situations consumers look for information on other platforms and/or give it different weights, but this should be investigated in the future. As well as whether eventual changes persist or not after their causes disappear.

In terms of type message design, people in Jordan would prefer more conversational messages than other types of messages, while people in Poland would prefer more scientific descriptive appeal one. Both nations, when assessing the degree of importance of the type of the educational awareness message concerning the coronavirus virus epidemic through electronic social media platforms, placed cartoons panic appeal and humorous appeal in the last two places. It can be assumed that in the case of problems of high importance for health, they rely to a greater extent on messages that are perceived in a more serious way and humorous messages, or cartoons associated with "children's" themes are not perceived by them as authoritative.

As the most important type of news, Poles pointed to scientific descriptive appeal, which in the opinion of Jordanians came only third, after conversational appeal and martyrdom appeal. An attempt to interpret such a result can be made by referring to the Inglehart-Welzel World Cultural Map (2020). According to one of the dimensions included in it, i.e., traditional values versus secular-rational values, Poland represents a culture that is significantly more rational than Jordan, which may explain the greater effectiveness of scientific messages. Jordan's characteristic adherence to traditional values for which respect for authority is related may also explain the greatest importance of senior state health officials as a source of messages.

In addition, it may indicate trust in government institutions and sources of information, as opposed to media health professionals/reporters who were assigned the last place by Jordanian respondents. Interestingly, Poles indicated exactly the opposite importance of the sources: the most important in their perception were media health professionals/reporters, and the least important—senior state health officials. This may be related to a negative assessment by 58% of Poles of the way their government is dealing with the crisis caused by the pandemic and suspicions that the government is exaggerating the pandemic crisis (51%) or hiding some information from the public (63%) (Rosa, 2020—research conducted in seven countries by the international organization in cooperation with the Kantar research centre). Moreover, in general, the confidence of Poles towards the government is low, in this respect it was ranked 6th from the end among OECD countries in 2017, [69].

## 6. Conclusions, Theoretical Contributions and Implications

This study fills the literature gap regarding a comprehensive understanding of a social media campaign (the type of social media platforms, the type of message appeal and the sources of the message sender) during COVID-19 as a global pandemic across national countries. It also significantly contributes to supporting the traditional communication theory in which claims that success of promotional health program is based on choosing the appropriate type of communication tool (i.e., social media platforms), type of message appeal and type of message (preferred sender) that relevant the target audiences' media habits and culture. The present study provides theoretical contributions to the literature on social media's role in public health care and behavioral change across national countries, including validating the research on another environmental culture context.

There are also significant implications from this study's research findings for governmental health officials, health professional care and practitioners, and other decision-makers in health organizations. First, they should be fully aware of the importance of launching a health promotional campaign in public health awareness and behavioral health change to adequately protect their community and nation from spreading pandemic diseases such as coronavirus (COVID-19). Such informed campaigns are all the more important as they must counterbalance the spread of disinformation and information from dubious sources available on social media platforms, as evidenced by recent studies in the context of the coronavirus pandemic [70]. Research conducted in Poland showed that the perceived risk of contracting COVID-19 and the actual risk was not related to each other, which was summarized by their authors "...media do not only reflect reality, but also create it" [71]. Second, they should consider the linkage between the health promotional campaign components (the type of medial social platform, the type of message appeal and the sources of the message) and target audiences' media habits and culture in any promotional health program future.

In conclusion, effective and timely health communication is always essential, and this works via public health authorities use of social media platforms with an appropriate type of message and an appropriate source of message. Knowing the target's media habits and culture is essential for effective promotional health campaign plans to raise awareness of public health and behavioral change during a global pandemic. Public health authorities must continue their efforts to raise public health awareness by disseminating brief messages to targeted populations. However, sometimes it is crucial to reach every citizen in society.

It may be relevant for everyone to act on an imminent pandemic. Some significant health efforts may depend on the audience's participation that cannot be accessed directly through media channels. More research is needed to validate how public promotional campaigns can be developed and launched to improve health knowledge and adopt healthy behaviors in the cross-cultural context.

**Author Contributions:** Data curation, R.A.-D.; methodology, H.A.-D.; resources, A.S. and M.B.-K.; writing—original draft, M.Y.A.; writing—review and editing, R.M. and Q.N.A. All authors have read and agreed to the published version of the manuscript.

**Funding:** This research received no external funding.

**Institutional Review Board Statement:** Not applicable.

**Informed Consent Statement:** PlWe confirm that all methods were carried out in accordance with relevant guidelines and regulations. We confirm that all experimental protocols were approved by the University of Jordan. We also confirmed that informed consent was obtained from all subjects or, if subjects are under 18, from a parent and/or legal guardian.

**Data Availability Statement:** Materials and data used in this literature review may be obtained from the first author.

**Acknowledgments:** We would like to express our deep and sincere gratitude to Princess Sumaya University for Technology (PSUT) and the University of Jordan for their support and for providing the research team with all the necessary facilities.

**Conflicts of Interest:** The authors declare no conflict of interest.

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
