# Peer review of "The Role of Social Media in Raising Public Health Awareness during the Pandemic COVID-19: An International Comparative Study"

_informatics, doi:10.3390/informatics8040080_

Round 1
Reviewer 1 Report
Referee report for the manuscript entitled “The Role of Social Media in Raising Public Health Awareness during Pandemic Covid-19: An International Comparative Study”
The manuscript focuses on an interesting topic—i.e., social media and public health—that is relevant to the journal and academic and professional audiences. While the manuscript is in general well written, it has some limitations to be considered for publication in this leading journal of the field. These issues are listed below.
- Abstract is structured in a weird way; bold highlighted parts should be removed from the abstract.
- The following highly relevant literature should be incorporated to consolidate the literature background of the manuscript.
- Andersen, K. N., Medaglia, R., & Henriksen, H. Z. (2012). Social media in public health care: Impact domain propositions. Government information quarterly, 29(4), 462-469.
- Dredze, M. (2012). How social media will change public health. IEEE Intelligent Systems, 27(4), 81-84.
- Schillinger, D., Chittamuru, D., & Ramírez, A. S. (2020). From “infodemics” to health promotion: a novel framework for the role of social media in public health. American journal of public health, 110(9), 1393-1396.
- Yigitcanlar, T., Kankanamge, N., Preston, A., Gill, P. Rezayee, M., Ostadnia, M., Xia, B., & Ioppolo, G., (2020). How can social media analytics assist authorities in pandemic related policy decisions? Insights from Australian states and territories. Health Information Science and Systems, 8(1), 37.
- Yigitcanlar, T., Kankanamge, N., Inkinen, T., Butler, L., Preston, A., Rezayee, M., Ostadnia, M., Gill, P., Ioppolo, G., & Senevirathne, M., (2021). Pandemic vulnerability knowledge visualisation for strategic decision-making: a COVID-19 index for government response in Australia. Management Decision, https://doi.org/10.1108/MD-11-2020-1527.
- Zhang, J., Chen, Y., Zhao, Y., Wolfram, D., & Ma, F. (2020). Public health and social media: A study of Zika virus‐related posts on Yahoo! Answers. Journal of the Association for Information Science and Technology, 71(3), 282-299.
- Discussion and Conclusion section can be improved to better address the so what question.
- References and reference list should follow journal’s guides.
Author Response
Dear Sir/Madam,
The authors appreciate the great efforts made by the reviewers and editor. We have carefully compiled all the comments to rewrite and resubmit the paper. The details of the point-by-point revisions are described as follows. The authors are also grateful to the anonymous reviewers for their thoughtful comments and suggestions, which have elevated the quality of this paper. Thank you all for your valuable comments. We appreciate your efforts and contributions.
Reviewer 1
The manuscript focuses on an interesting topic—i.e., social media and public health—that is relevant to the journal and academic and professional audiences. While the manuscript is in general well written, it has some limitations to be considered for publication in this leading journal of the field. These issues are listed below.
- Abstract is structured in a weird way; bold highlighted parts should be removed from the abstract.
Bold highlighted parts are removed and the abstract has been adjusted according to the guidelines.
- The following highly relevant literature should be incorporated to consolidate the literature background of the manuscript.
We have incorporated valuable references into our study.
- Andersen, K. N., Medaglia, R., & Henriksen, H. Z. (2012). Social media in public health care: Impact domain propositions. Government information quarterly, 29(4), 462-469.
- Dredze, M. (2012). How social media will change public health. IEEE Intelligent Systems, 27(4), 81-84.
- Schillinger, D., Chittamuru, D., & Ramírez, A. S. (2020). From “infodemics” to health promotion: a novel framework for the role of social media in public health. American journal of public health, 110(9), 1393-1396. Not in English
- Yigitcanlar, T., Kankanamge, N., Preston, A., Gill, P. Rezayee, M., Ostadnia, M., Xia, B., & Ioppolo, G., (2020). How can social media analytics assist authorities in pandemic related policy decisions? Insights from Australian states and territories. Health Information Science and Systems, 8(1), 37.
- Yigitcanlar, T., Kankanamge, N., Inkinen, T., Butler, L., Preston, A., Rezayee, M., Ostadnia, M., Gill, P., Ioppolo, G., & Senevirathne, M., (2021). Pandemic vulnerability knowledge visualisation for strategic decision-making: a COVID-19 index for government response in Australia. Management Decision, https://doi.org/10.1108/MD-11-2020-1527.
- Zhang, J., Chen, Y., Zhao, Y., Wolfram, D., & Ma, F. (2020). Public health and social media: A study of Zika virus‐related posts on Yahoo! Answers. Journal of the Association for Information Science and Technology, 71(3), 282-299.
- Discussion and Conclusion section can be improved to better address the so what question.
Thank you for your valuable comments, we considered them; thus, we enhanced the manuscript.
- References and reference list should follow journal’s guides.
We followed the journal’s guidelines.
Reviewer 2 Report
Dear authors, the article has an interesting approach but I must reject the publication of this article as it does not meet the inclusion criteria. There are other journals in which it could be published, for example, EJIHPE.
Furthermore, it has major shortcomings that should be corrected and that are not in accordance with the quality that an article should have to be published.
I outline some of these shortcomings:
- Check the references, they are not numbered according to their appearance in the article. They are also ordered alphabetically, rather than by appearance.
- The abstract exceeds the recommended 200 words.
- COVID is presented in upper case (e.g. lines 367 and 371) and in lower case (e.g. lines 530, 539 and 553, among others), the presentation should be homogenised. Line 511 is written Covide.
- In line 434 Cranach's is written instead of Cronbach.
- There are numerous widowed words at the end of sentences, as in lines 27, 108, 129, 133, 138, etc.
- In the acknowledgements section, the template appears, but not your acknowledgements.
- The paragraphs are made up of short sentences with references, without contributing any literature of their own, thus, between lines 98 and 106 we find 5 quotations and 5 sentences, and no personal contribution.
There are deficiencies in the methodology, where material and method are not properly explained.
- Inclusion and exclusion criteria, the universe, and the population are not stated.
- How was the sample selected?
- There is a lack of clarity in the definition of variables (there may be confounding variables) and working hypotheses.
- The tables in the Discussion should be included in the analysis.
- ANOVA is a parametric analysis. Has normality and homoscedasticity been checked? What tests have been used?
- What regression analyses have been used?
- How have the variables been chosen, e.g. the type of preferred message design?
The analysis is confusing and carries over into the conclusions, and these sections need to be redrafted to make them clearer to the reader.
I encourage you to redo it and send it as the topic is interesting and good conclusions can come out of the study concerning the use of social networks in public health policies.
Best regards.
Author Response
Reviewer 2
Dear Sir/Madam,
The authors appreciate the great efforts made by the reviewers and editor. We have carefully compiled all the comments to rewrite and resubmit the paper. The details of the point-by-point revisions are described as follows. The authors are also grateful to the anonymous reviewers for their thoughtful comments and suggestions, which have elevated the quality of this paper. Thank you all for your valuable comments. We appreciate your efforts and contributions.
Furthermore, it has major shortcomings that should be corrected and that are not in accordance with the quality that an article should have to be published.
I outline some of these shortcomings:
- Check the references, they are not numbered according to their appearance in the article. They are also ordered alphabetically, rather than by appearance.
We have numbered references according to appearance.
- The abstract exceeds the recommended 200 words.
Abstract has been adjusted to the proper guideline.
- COVID is presented in upper case (e.g. lines 367 and 371) and in lower case (e.g. lines 530, 539 and 553, among others), the presentation should be homogenised. Line 511 is written Covide.
We have made the word COVID homogenized in the entire documents.
- In line 434 Cranach's is written instead of Cronbach.
Cranach's has been replaced with Cronbach's.
- There are numerous widowed words at the end of sentences, as in lines 27, 108, 129, 133, 138, etc.
- In the acknowledgements section, the template appears, but not your acknowledgements.
The acknowledgments section has been added.
- The paragraphs are made up of short sentences with references, without contributing any literature of their own, thus, between lines 98 and 106 we find 5 quotations and 5 sentences, and no personal contribution.
The personal contribution has been made clear.
There are deficiencies in the methodology, where material and method are not properly explained.
- Inclusion and exclusion criteria, the universe, and the population are not stated.
- How was the sample selected?
- There is a lack of clarity in the definition of variables (there may be confounding variables) and working hypotheses.
- The tables in the Discussion should be included in the analysis.
- ANOVA is a parametric analysis. Has normality and homoscedasticity been checked? What tests have been used?
- What regression analyses have been used?
- How have the variables been chosen, e.g. the type of preferred message design?
The analysis is confusing and carries over into the conclusions, and these sections need to be redrafted to make them clearer to the reader.
- Methodology
The research utilizes a quantitative method with an exploratory and descriptive design. To confirm the research's conceptual model and to examine the research hypotheses, a survey questionnaire was executed to collect it. This research's target population consisted of all users of mass media channels in Jordan, and Poland. A convenience sampling technique was employed. Consequently, a sample size of 1149 respondents was reached. The questionnaire consisted of two sections; the first section in questionnaire presents general personal information about a respondent, gender, age, educational level, and country. The second section includes questions to measure the independent and dependent variables. The questionnaire's content was mainly selected and adopted from relevant previous studies [64] using a 5-points Likert scale ranging from strongly disagree =1 to strongly agree =5 to measure four independent variables namely social media network, Public Awareness, Behavioral Change, and Public Protection. In contrast, the dependent variables, namely Type of Message Appeal and Source of the Message, were measured using-points Likert scale ranging from not necessary at all =1 to very important=5. A web link for the online questionnaire was sent out to the potential respondents between May 15th to Nov.30th in 2020.
Reviewer 3 Report
Interesting topic and timely. I was particularly found of figure 1. Method selected is appropriate and easy to follow analysis.
Table 4, last column VIP to VIF
Line 50, you may perhaps find additional sources that states the spread of the covid in the media being within 24 hrs or a week later when people realized this was serious.
Author Response
Reviewer 3
The authors appreciate the great efforts made by the reviewers and editor. We have carefully compiled all the comments to rewrite and resubmit the paper. The details of the point-by-point revisions are described as follows. The authors are also grateful to the anonymous reviewers for their thoughtful comments and suggestions, which have elevated the quality of this paper. Thank you all for your valuable comments. We appreciate your efforts and contributions.
Interesting topic and timely. I was particularly found of figure 1. Method selected is appropriate and easy to follow analysis.
Table 4, last column VIP to VIF
We have corrected the column from VIP TO VIF
Line 50, you may perhaps find additional sources that state the spread of the covid in the media being within 24 hrs or a week later when people realized this was serious.
Molla (2020) has mentioned the information in this study “In just 24 hours, there were 19 million mentions of COVID-19 across social media and news sites worldwide”
Reviewer 4 Report
- This is a significant research by employing the quantitative research method of distributing survey questionnaires to explore the role of social media in raising public health awareness during pandemic Covid-19 in the two selected countries.
- It could be better if there is a clear definition of “social media campaign”, and also, we need clearer explanations on why the research explores the research questions from the following perspectives: the type of platforms, preferred message design/types, and preferred source of message.
- As to the research method, it is fine for the research to adopt the quantitative method by distributing survey questionnaires. However, individual experience is very important in examining the research questions in this research. Maybe a supplement of interview with the social media users will give an authentic picture of the social media users and can provide a sound explanation to the quantitative findings.
- It’s good that the research developed a conceptual framework based on a communication literature and belief health change theory. However, public awareness and behavioural change involve in psychological factors. It’ll be better if the research questions are explored by employing psychological explanations.
- Please refine the sentences, and pay attention to the grammar and spelling mistakes. For example, Grammar mistake to the sentence on p2. “Public awareness and prevention of Coronavirus(COVID-19).”Spelling mistakes such as “youthpeople”on p4 should be modified. Grammar mistakes in the sentence on p4:“Nowadays, a different method is suggested according to previously valuationof social preferences.”The sentence---“Infection diseases play an important role in disease control; …”on p2 is not so clear to indicate the ideas that the authors would like to convey.
Author Response
Reviewer 4
The authors appreciate the great efforts made by the reviewers and editor. We have carefully compiled all the comments to rewrite and resubmit the paper. The details of the point-by-point revisions are described as follows. The authors are also grateful to the anonymous reviewers for their thoughtful comments and suggestions, which have elevated the quality of this paper. Thank you all for your valuable comments. We appreciate your efforts and contributions.
- This is a significant research by employing the quantitative research method of distributing survey questionnaires to explore the role of social media in raising public health awareness during pandemic Covid-19 in the two selected countries.
Thank you, we appreciate your efforts and contributions.
- It could be better if there is a clear definition of “social media campaign”, and also, we need clearer explanations on why the research explores the research questions from the following perspectives: the type of platforms, preferred message design/types, and preferred source of the message.
Thank you for your valuable comments, we considered them; thus, we enhanced the manuscript.
- As to the research method, it is fine for the research to adopt the quantitative method by distributing survey questionnaires. However, individual experience is very important in examining the research questions in this research. Maybe a supplement of interview with the social media users will give an authentic picture of the social media users and can provide a sound explanation to the quantitative findings.
It was very difficult to interview people during corona; for that reason, the author focused on quantitative methods.
- It’s good that the research developed a conceptual framework based on a communication literature and belief health change theory. However, public awareness and behavioural change involve in psychological factors. It’ll be better if the research questions are explored by employing psychological explanations.
Thank you for your valuable comments, we considered them; thus, we worked on the quality of writing.
- Please refine the sentences, and pay attention to the grammar and spelling mistakes. For example, Grammar mistake to the sentence on p2. “Public awareness and prevention of Coronavirus(COVID-19).”Spelling mistakes such as “youthpeople”on p4 should be modified. Grammar mistakes in the sentence on p4:“Nowadays, a different method is suggested according to previously valuationof social preferences.”The sentence---“Infection diseases play an important role in disease control; …”on p2 is not so clear to indicate the ideas that the authors would like to convey.
Thank you for your valuable comments, we considered them; thus, we worked on the quality of writing.
Round 2
Reviewer 2 Report
I would like to thank the authors for their efforts to adapt the article to the reviewers' suggestions. Thank you for the opportunity to revisit this paper.
The paper is more complete and makes it easier to read and understand.
Great work. Regards.